# Language Through a Prism: A Spectral Approach for Multiscale Language Representations

**Alex Tamkin**[†]
Stanford University

**Dan Jurafsky**
Stanford University

**Noah Goodman**
Stanford University

## Abstract

Language exhibits structure at different scales, ranging from subwords to words, sentences, paragraphs, and documents. To what extent do deep models capture information at these scales, and can we force them to better capture structure across this hierarchy? We approach this question by focusing on individual neurons, analyzing the behavior of their activations at different timescales. We show that signal processing provides a natural framework for separating structure across scales, enabling us to 1) disentangle scale-specific information in existing embeddings and 2) train models to learn more about particular scales. Concretely, we apply spectral filters to the activations of a neuron across an input, producing *filtered embeddings* that perform well on part of speech tagging (word-level), dialog speech acts classification (utterance-level), or topic classification (document-level), while performing poorly on the other tasks. We also present a *prism layer* for training models, which uses spectral filters to constrain different neurons to model structure at different scales. Our proposed BERT + Prism model can better predict masked tokens using long-range context and produces multiscale representations that perform better at utterance- and document-level tasks. Our methods are general and readily applicable to other domains besides language, such as images, audio, and video.

## 1 Introduction

Language exhibits structure at multiple levels, ranging from morphology at the subword level [1], word meaning at the lexical level [2], coherence and other discourse properties at the clause or sentence level [3, 4, 5], to topical and narrative structures for entire documents [6, 7]. Prior work in NLP has shown how these kinds of structures can be explicitly modeled by representing individual levels of structure [8, 9, 10, 11, 12, 13], multiple levels of structure [14, 15, 16], building hierarchical models that capture structure at the sentence level [17, 18] or between sentences [19, 20], and probing to discover known linguistic levels of structure [21, 22, 23, 24].

We propose a new method for uncovering and learning this kind of structure in representations at every scale, from word meaning to document topics, without drawing on prior linguistic models of specific structural levels like "sentence" or "clause." To do so, we employ tools from spectral analysis, widely used in signal processing and other fields [25] to separate and control information at different timescales. Intuitively, any sequence of values, such as a neuron's activations across input tokens, can be represented as a weighted sum of cosine waves with different frequencies. The weight for a particular frequency indicates the amount of structure in the sequence at that scale: weight on higher frequencies indicates faster changes in the neuron's activation from token to token, while weight on lower frequencies indicates activations that shift more gradually across an input. By removing certain

---

[†]atamkin@stanford.edu

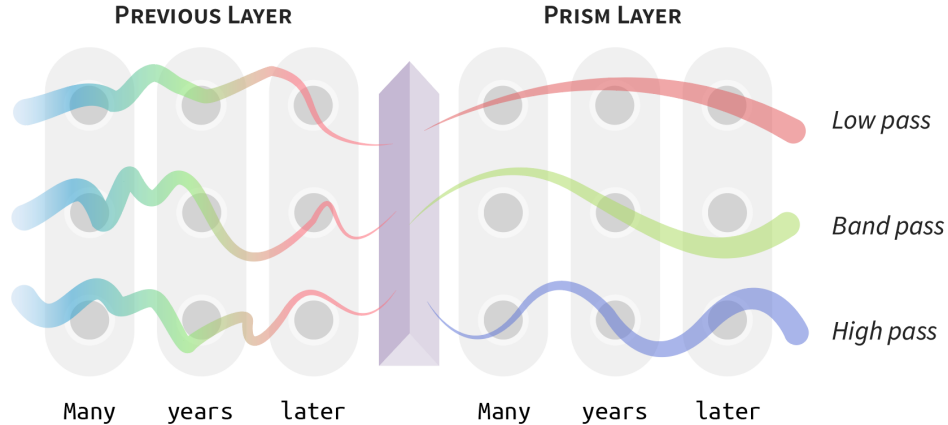

Figure 1: **The prism layer specializes different neurons for different scales.** First, the representations for an input are computed (left; in this case, the input is of length three). Next, a spectral filter (a low-, high-, or band-pass) is applied along the activations of each individual neuron (right). This produces neurons that are only able to represent structure at particular scales. Curved lines illustrate the scales at which neurons can change over an input.

frequencies, called *spectral filtering*, we can remove information about variation at particular scales. See Figure 2 for a visualization.

In this work, we apply spectral filters to the activations of individual neurons in BERT [26], a popular deep NLP model. This enables us to separate information in model representations that changes at different rates across the input—for example, part of speech changes on a word-to-word basis, while topical changes are much more gradual. Concretely, we contribute:

1. **A principled framework** based on spectral analysis for describing structure at multiple scales in deep representations. While we consider applications to NLP models, this is a general framework that could extend to other models with representations arranged in spatial or temporal structure. (Section 2)

2. **A technique**, *spectral filtering*, for extracting scale-specific information from language representations. We show how low-pass filters can alter representations to only perform well on topic classification (document-level), while band-pass and high-pass filters do the same for dialog acts classification (utterance-level) and part of speech tagging (word-level). (Section 3)

3. **A new model component**, the *prism layer*, which specializes neurons in a model for particular scales of structure. After training with a prism layer, our model is more sensitive to long-range interactions between tokens and produces individual representations that perform comparably or better than BERT's across tasks at different scales. (Section 4)

## 2 Spectral filtering of contextual word representations

This section provides some background on the spectral analysis tools we use and describes how we apply them to deep language representations.

### 2.1 Background: The discrete cosine transform and spectral filters

In order to perform operations in the frequency domain of a sequence, we first need to obtain a representation of the input in the frequency domain. This is the role of a *spectral transform*. The spectral transform we use in this work is the discrete cosine transform (DCT[2]) [27], a widespread tool used in audio coding, texture analysis, image classification, and compression [28, 27]. The DCT represents a real-valued sequence of points as a same-length sequence of weights over cosine

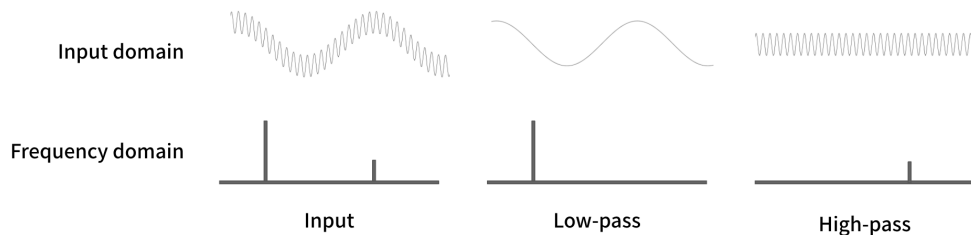

Figure 2: **A visual depiction of spectral filters and their effects in the input and frequency domain.** The input domain shows a sequence of values (e.g., the activation of a neuron across input tokens). The frequency domain shows the weight on the cosine waves which sum to produce the curve in the input domain. Low-pass filters only allow low frequencies to pass through, producing a smoothed input. High-pass filters only allow high frequencies and produce a locally-normalized input. Band-pass filters (not shown) are compositions of low- and high-pass filters.

functions of different frequencies. Formally, for a real-valued sequence $\{x^{(0)} \dots x^{(N-1)}\}$ its DCT (the weights for each frequency) is obtained by

$$f^{(k)} = \sum_{n=0}^{N-1} x^{(n)} \cos \left[ \frac{\pi}{N} \left( n + \frac{1}{2} \right) k \right] \qquad k = 0, \dots, N-1 \qquad (1)$$

Intuitively, the DCT computes the similarity of a signal and cosine waves of different frequencies by taking the dot product between them. These dot products constitute the coefficients of the signal in the frequency domain. The DCT is closely related to the discrete Fourier transform (DFT). We use the DCT here because it is a real-to-real function (the DFT is complex-to-complex), is widely used in practice, and can often produce fewer artifacts than the DFT when filtering [27, 29].

The DCT of a sequence enables straightforward manipulation of structure at different scales in a sequence. For example, one can remove components above some threshold frequency $k_{\text{thresh}}$ by setting $f_k \leftarrow 0$ for all $k > k_{\text{thresh}}$, then applying the inverse DCT (IDCT) to return the signal to the original domain [30]. This is known as a *low-pass filter*, and returns a smoothed, same-length version of the original input, removing shorter-term fluctuations. The inverse operation can be performed to achieve a *high-pass filter*, which returns a signal where each term is locally normalized with respect to its neighbors, neutralizing longer-term trends. Composing these two operations yields a *band-pass filter*, as only a band of frequencies is allowed to pass through the filter. See Figure 2 for a visual depiction.[3]

## 2.2 Applying the DCT to contextual word representations

How do we apply the DCT to deep language representations? A common feature of modern NLP models is *contextual word representations*, a sequence of vectors created by processing a sequence of tokens (e.g., words or subword units). These representations are produced by a wide range of modern NLP architectures, including Transformer-based [34] models like BERT [26] and GPT-2 [35], as well as LSTM-based [36] models such as ELMo [37].

Assume we are given a sequence of contextual word representations $v_0, \dots, v_{N-1}$. The core technique we propose is to apply the DCT to a slice of these representations *along a single neuron*: $v_0[i], \dots, v_{N-1}[i]$. We refer to the transformed sequence in the frequency domain $f_0[i], \dots, f_{N-1}[i]$ as the spectrum of the $i$th neuron. $f_0[i]$ is the lowest frequency term, corresponding to the average value of $v_0[i], \dots, v_{N-1}[i]$, while $f_{N-1}[i]$ is the highest frequency term. We can then implement any of the filters from Section 2.1 by zeroing out the appropriate values in the spectrum, and then applying the IDCT to return the sequence to the original domain. In practice, external libraries make this quite simple: we show a three-line implementation of a low-pass filter in Figure 3b.

| Filter | Ex. Scale | Period (toks) | DCT index |
|---|---|---|---|
| HIGH | Word | 1–2 | 130–511 |
| MID-HIGH | Clause | 2–8 | 34–129 |
| MID | Sentence | 8–32 | 9–33 |
| MID-LOW | Paragraph | 32–256 | 2–8 |
| LOW | Document | 256–∞ | 0–1 |

(a) **The spectral filters we consider in this work**, along with their periods, spectral bands (the indices in the DCT), and example linguistic phenomena at that scale. The period of a cosine wave for a DCT index is the approximate number of tokens it takes for the wave to complete a cycle.

```
def low_pass(H, k):
    H_dct = dct(H.T)
    H_dct[:, k:] = 0
    return idct(H_dct).T
```

(b) **Spectral filters are simple to incorporate into existing models.** Python-style code for a low-pass filter over representations. Input H is a list of representations for each input token, while k is the low-pass threshold frequency. T is the transpose operator. We use a PyTorch library to compute the (I)DCT.

Figure 3

## 3  The relationship between spectral frequencies and linguistic phenomena

We have seen how to apply spectral filters to the hidden states of deep NLP models. In this section, we explore how these spectral filters can be used to separate out phenomena at different scales in contextual word representations.

### 3.1  Disentangling scale-specific information in representations

Contextual word representations have been shown to not only encode the meaning of tokens in context [37], but also a wider range of linguistic phenomena such as semantic roles, entity types, constituent labels, relations between entities, and coreference [38]. This suggests that these representations may already be encoding information about multiple scales ranging from the (sub)word itself to its containing phrase, clause, sentence, paragraph and perhaps the document as a whole. In this work, we consider whether these phenomena can be separated out at the level of *individual neurons* by using spectral filters to tease apart structure at different scales in a neuron's activations across an input.

To investigate, we observe how the choice of spectral filter affects the ability of a classifier to perform tasks at different scales using the filtered representations. Each spectral filter is determined by a corresponding *spectral band*: the range of frequencies that is used for the low-, high-, or band-pass. We seek to choose bands corresponding to different scales. However, the scale of a particular frequency is revealed by its period: the number of tokens it takes to complete a full cycle. For example, from Equation 1 we see that index $8$ of the DCT has a frequency of $2 * 8 = 16$, and thus for inputs of size $512$ has a period of $512/16 = 32$ tokens.

In this work, we divide the frequency spectrum into five bands, chosen reflect the inductive bias that linguistic units at one scale are composed of multiple units from the scale below (e.g. several words compose a phrase). Thus, we allocate bands such that for each band, the periods of the frequencies in the next higher band decay by a fixed amount. This produces five bands (LOW, MID-LOW, MID, MID-HIGH, and HIGH) with a diverse range of scales, as shown in Table 3a.[4] See the Appendix for more details on band allocation and discretization.

### 3.2  Probing bandpassed representations for linguistic information

We evaluate the content of these filtered representations through probing experiments [39, 40, 41]. For each dataset below, we encode each training example with a fixed, pretrained BERT-Base cased model [26]. This produces a series of 768-dimensional contextual word representations. We then apply a spectral filter along each dimension and train a softmax classifier to perform a particular task using each filtered representation. We examine three English-language tasks, involving classification of word-, utterance-, and document-level phenomena, providing a natural testbed for investigating the content of these representations:

1. **Part of speech tagging (word-level):** We use the Penn Treebank dataset [42]. The task is to predict the part of speech (e.g. PAST TENSE VERB, WH-PRONOUN, CARDINAL NUMBER) from the given token representation.

2. **Dialog speech act classification (utterance-level):** We use the Switchboard Dialog Speech Acts corpus [43, 44, 45].[5] The task is to predict the dialog speech act (e.g. APOLOGY, HEDGE, APPRECIATION) of the utterance containing the given token representation.

3. **Topic classification (document-level):** We use the 20 Newsgroups dataset [46]. The task is to predict the topic (newsgroup; e.g. SCI.SPACE, COMP.GRAPHICS, REC.AUTOS) of the document containing the given token representation.

We train our probing models for a maximum of 30 epochs, using the Adam optimizer [47] with default parameters. We use early stopping with a patience of one, decaying the learning rate by a factor of 2 when successive epochs do not produce a decrease in validation loss. To compare against the masked language modeling (MLM) task, which was the original target task[6] for these representations [26], we also train an MLM probe for three epochs on the WikiText-103 dataset [48].

As Figure 4 shows, different spectral filters indeed produce representations specialized for the expected task. The highest probing accuracy for part of speech tagging occurs when extracting the HIGH band, aligning with the fact that this is a word-level task. However, the highest frequency spectral band still performs worse than the original representations, suggesting that lower frequency information is sometimes necessary for this task (e.g. for parts of speech correlated over several tokens, such as strings of numbers or lists of nouns). By contrast, topic-classification performs best with information from the LOW band, aligning with the fact that it is a document-level phenomenon. Interestingly, the accuracy for the LOW band is substantially higher than for the original representations, suggesting that higher frequency variation present in the original representations may be harmful for that task. Meanwhile, probing for dialog speech acts, a classification task over utterances, is most successful at the MID band, with performance comparable to that of the original representations. The probing results for masked language modeling are most similar to part of speech tagging, underscoring the degree to which MLM is a local task.

These results demonstrate that spectral filters are effective tools for separating multiscale linguistic phenomena in contextual word representations.

## 4 Using spectral filters during training

In the previous section, we saw how spectral filters can be used to isolate information about linguistic phenomena at different scales in an existing model's representations. However, this observed structure arose naturally from BERT's masked language modeling task, which we saw is a relatively local task. In this section, we will show how spectral filters can be used *during training* to produce multiscale representations with improved performance on mid-scale and global tasks despite being trained with masked language modeling.

### 4.1 The prism layer

In BERT, the information for the different tasks discussed above may be distributed across all neurons, rather than specialized in particular ones. Spectral filters, however, provide a natural way to force BERT to use different neurons for information about different scales. The resulting multiscale representations may then be better suited for a broader range of tasks than the original BERT representations.

To accomplish this, we take a given hidden state in BERT and divide the units evenly into five *sectors*.[7] To each sector, we then apply a different band-pass from Table 3a. We call these additional computations a *prism layer*, as they separate out the different frequencies in a layer's representations. See Figure 1 for an illustration. In our main experiments, we apply one prism layer after the last BERT layer. See the Appendix for an investigation of placing prism layers after each BERT layer.

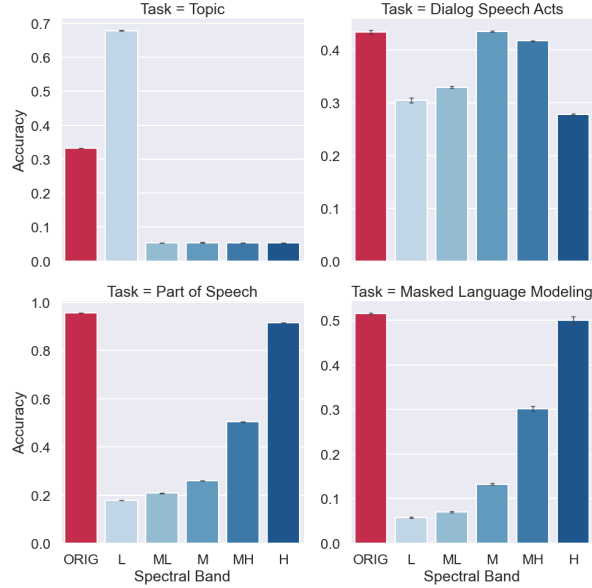

Figure 4: **Different spectral filters extract information useful for tasks at different scales.** Probing accuracy for different tasks and band-passes. A low-pass filter produces representations that yield highest probing accuracy on topic classification, while high-passed representations have highest probing accuracy for part of speech tagging. Meanwhile, band-passing the middle frequencies is most useful for dialog speech act probing. "ORIG" refers to the performance of the original token representations. Error bars show standard deviations over three probing runs.

We then train our pretrained BERT model with the prism layer on the masked language modeling task; this is so the model can adjust to the new constraints imposed upon it and learn to allocate information at particular frequencies to the correct sectors. We use an external PyTorch library for computing and backpropagating through the DCT and IDCT. [8] We train on the WikiText-103 dataset [48] for 50k steps at a batch size of 8 with default parameters for Adam. To allow for fair comparisons between our model and BERT, we also further train an unmodified pretrained BERT model using this same data and procedure (see the Appendix for an ablation of this step).

## 4.2 Results

We now compare the probing performance of the vanilla BERT model with the BERT model trained with our prism layer, shown in Table 1. The BERT model with the prism layer performs considerably better than BERT on topic (+18.8%) and dialog speech act (+6.9%) classification while maintaining high accuracy on part of speech tagging (-1.5%). These results demonstrate that the prism layer has enabled BERT to produce more general-purpose representations that capture phenomena across scales.

## 4.3 Sensitivity to distant tokens

The multiscale representations produced by the prism layer are used by the model to perform the masked language modeling (MLM) objective. Since these representations contain information at different scales, this provides an inductive bias for the model to rely on both long-range and short-range information when performing the MLM task. To show this quantitatively, we consider an MLM problem where one hundred consecutive tokens in the middle of the input are masked. The model's loss on these tokens reflects the model's ability to rely on distant information to predict tokens without local context.

We plot the average log probability of the correct token in Figure 5, for both the BERT model with the prism layer, as well as a BERT model trained on WikiText-103 for the same number of steps. As

Table 1: **Training with a prism layer produces multiscale representations that perform comparably or better than BERT across different tasks.** Probing accuracy and standard deviation (3 trials) for different tasks on the final-layer BERT and BERT + Prism representations.

| Task | Model | Accuracy (%) | S.D. (%) |
|---|---|---|---|
| Topic classification | BERT | 32.21 | 0.08 |
| | **BERT + Prism** | **51.01** | 0.14 |
| Dialog speech acts | BERT | 47.09 | 0.33 |
| | **BERT + Prism** | **54.02** | 0.61 |
| Part of speech | **BERT** | **95.86** | 0.02 |
| | BERT + Prism | 94.41 | 0.02 |

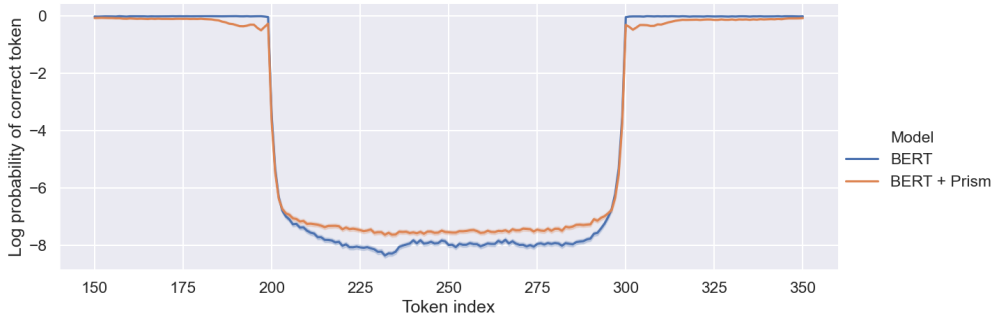

Figure 5: **Training with a prism layer significantly improves prediction of masked tokens without local context (note the log scale).** Average log probability of correct token for different indices (N=1600). Indices between 200 and 300 are replaced with a [MASK] token in the input, requiring the model to use long-range context to generate a probability distribution for the missing token. The higher log probabilities in the masked region for the BERT + Prism model suggest the prism layer makes the model more sensitive to long-range dependencies. Shaded regions are 95% bootstrap CIs (generally too small to see without magnification).

expected, no model can precisely guess the missing tokens with perfect accuracy. But we do see a noticeable difference between the probabilities assigned to the correct token by the BERT models with and without the prism layer (note the log scale). This indicates that the BERT + Prism model is a better *long range* language model, using context to predict distant tokens.

Another interesting phenomenon in the graph is the dips in log probability exhibited by the BERT + Prism model adjacent to the redacted text, indicating dependence on (redacted) distant context. No such dip exists for the original BERT model, indicating that it solves the MLM task in a very local way. These results suggest that the prism layer is a useful tool for encouraging modeling of long-range dependencies in Transformer models [49, 36].

## 5   Related work

Our work connects with several streams of research investigating multiscale structure in natural language and our models of it. Prior work has studied the extent of this structure at different scales in linguistic corpora, using tools ranging from random walk models and power spectra [50, 51] to entropy and mutual information [52]. To model this structure, researchers have conducted multiresolution analyses of text corpora by applying diffusion wavelets to term-document corpora [53], multinomial topic distributions [54], and term-term cooccurrence graphs [55]. Concerning deep learning, several works have considered the challenges of modeling different scales in distributed representations of words [8, 56] and of capturing long-term dependencies in recurrent neural networks [36, 57]. Other work conducts analytic studies of models that illuminate their scale-awareness, including the sensitivity of LSTM language models to relationships at different scales [58] and the attention patterns

of Transformer models [59]. In conversation with this literature, our work provides a principled way of understanding multiscale structure in the representations of deep models, illuminating the linguistic phenomena captured at each of these scales and enabling the construction of scale-specific representations for downstream purposes.

In concert with these analyses, a large body of work has attempted to leverage the expressive capability of distributed representations to improve modeling at particular scales. For example, several works introduce different kinds of architectural modifications to recurrent neural networks in order to encourage learning hierarchical structure, especially long-term structure, including via updating hidden states at different intervals [60], multilayered models [61, 62], incorporating tree structures [17] or syntactic parsing [18], introducing residual connections [63], adding auxiliary losses [49], and discretizing ordinary differential equations [64]. In addition, certain works explicitly focus on creating high-quality representations at particular scales, including the word-level [8, 9], sentence-level [10, 11, 65, 20], paragraph-level [12] and document-level [13]. Perhaps most similar to our work is a stream of work incorporating the Fourier basis into recurrent architectures [66, 67]. However, while these works focus on speeding up training or improving gradient flow in RNNs, our approach is architecture-agnostic, provided the model produces contextual word representations, and can be used to understand or improve specific scales of interest in the model's representations. Another piece of related work is Ordered Neurons [68], which enforces an update hierarchy in the latent state of an RNN to capture tree-like structure in an input (e.g., syntax trees). By comparison, our approach generalizes beyond RNN or autoregressive architectures and can capture both syntactic structure like part of speech as well as longer-range multiscale phenomena like dialog speech acts and topic where tree structures may not be as appropriate.

Finally, our work is related to spectral approaches in audio [69, 70], where it is naturally suited as an input representation, as well as computer vision, where the Fast Fourier Transform [71] and the Discrete Cosine Transform [30] have been used to speed up the training of convolutional neural networks [72], generate filters for scene classification [73], and compress convolutional models [74]. Concerning scales, the authors of StyleGAN [75] investigate how different layers in their model are responsible for phenomena at different scales, such as pose, lighting, face shape, and finer facial features. Most related to our work is a line of research that improves training by using spectral filters to replace downsampling operations in convolutional models [76] as well as improving optimization speed and generalization by removing low-magnitude [77] or high-frequency [76] spectral coefficients. We also explore attenuation of different frequency coefficients, but in an NLP context to improve modeling of long-range dependencies, and further use spectral techniques to understand, control, and improve modeling at different scales.

# 6   Conclusion

In this work, we demonstrate how techniques from spectral analysis provide a principled and effective framework for separating multiscale phenomena in deep language representations. We first demonstrate how spectral filters can be used to separate information at different scales in BERT representations. We use this technique to produce scale-disentangled representations that perform well at either part of speech tagging, dialog acts classification, or topic classification, while performing poorly on the other two tasks. We also show how to create multiscale representations by training with a prism layer, which forces different neurons to capture information about different scales. The representations produced by the resulting model enable comparable or higher performance across the three tasks than vanilla BERT representations. We also show that training with a prism layer increases the model's sensitivity to long-range context, as measured by a masked language modeling task. These results demonstrate that spectral techniques are a powerful set of tools for uncovering and modeling multiscale phenomena in deep NLP models.

Our work provides multiple avenues for further study. For interpretability researchers, these tools could enable better understanding of knowledge and information processing at different scales in neural models across different tasks, inputs, and layers. For researchers of linguistic change, this method may enable better tracking of topics over time or facilitate the removal of extraneous information (e.g., topic) when targeting a linguistic phenomenon at a different scale. Finally, we also see promise for improving NLP models during training in a broader range of applications and architectures. More generally, we emphasize that our method is domain agnostic: it needs only a collection of representations with some kind of geometric (e.g., spatial or temporal) structure—thus,

we are optimistic about the potential for further applications of these techniques on the hidden states of computer vision, time series, and reinforcement learning models, among others.

## Broader Impact

The spectral tools we provide in this paper are applicable to a wide range of neural network models and possible end uses. While this makes it difficult to speak with confidence about broader impacts of the research, we briefly discuss a few potential use cases. Scale isolation enables users to remove information about particular kinds of structure inside existing representations. This could be useful for interpretability or fairness research, as well as computational social scientists who wish to remove e.g. topical information from word embeddings. However, scale isolation may also enable tailored search for particular kinds of information in text or other content, which could enable uses that are beneficial or harmful depending on the use case and whether consent is obtained by relevant parties. The prism layer falls under a general trend of producing more capable neural networks. Such a trend may contribute to increased automation or other changes in labor markets, which may create benefits and harms that depend on the economic and social policies of relevant governing bodies.

## Acknowledgments and Disclosure of Funding

We would like to thank Shyamal Buch, Jesse Mu, Shikhar Murty, Ben Newman, Mike Wu, Pratyusha Ria Kalluri, and Jesse Michel for useful discussions and comments on drafts. This work was supported in part by DARPA under agreement FA8650-19-C-7923.

## Footnotes

[2]More precisely, this transform is known as the DCT-II

[3]Fully zeroing out frequencies (a *brick wall filter*) can produce artifacts after performing the IDCT, motivating the use of smoother attenuation functions [31, 32], which reduce artifacts in exchange for allowing less-than-full attenuation of frequencies outside the desired band. However, for simplicity, we use brick wall filters in this work, leaving study of other filters, as well as other spectral tools like wavelets [33], for future work.

[4]We use these five separate bands in part for instructive purposes; however, in practice, one might wish to smoothly change the endpoints of the spectral band across neurons. One could also specifically choose bands for a task based on their corresponding periods to include or exclude particular scales of interest.

[5]We use the preprocessing library from `https://github.com/cgpotts/swda`

[6]We do not consider the next sentence prediction task (NSP) [26]. While it was also used for BERT pretraining, we discard the [CLS] tokens, which are used to predict the NSP label.

[7]We distribute the $768 \pmod 5 = 3$ remaining units to the LOW, LOW-MID, and MID bands.

[8]https://github.com/zh217/torch-dct

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
