[Reviews · NeurIPS 2020]

Review 1

Summary and Contributions: This paper proposes to put spectral methods to use in modern NLP neural network models. The first use is that of extracting timescale specific representations, which represent information only at certain grains over sequences of tokens. The authors show that the extracted representations align with the appropriate timescale by showing that performance on tasks which mainly operate on a given timescale are best served by the corresponding representation. Second, the paper proposes a modeling technique to force increased timescale-organized behavior in the representations output by the model. Here, the authors show that the representations given this additional structure match or outperform the baseline on the tasks that correspond to the desired timescale. The paper concludes with some interesting analysis about how this kind of spectral module can increase sensitivity to long-term dependencies. I have read the rebuttal.

Strengths: 1. This paper is well-written and easy to understand. 2. The idea and techniques introduced are an innovative use of the familiar tool of spectral models and timescale analyses. 3. The results given are promising, showing both improvements on traditional metrics, but also the ability to selectively control these improvements, by choice of representation. 4. The analysis about long-term dependencies, though somewhat limited, presents a novel way of looking at usage of distant context.

Weaknesses: . There are relatively few experiments; with such a new idea, it is important to validate that it is not simply these datasets or even these tasks that contain this kind of structure. Since there are many tasks at all of these timescales, the lack of further results is somewhat disappointing. 2. The authors do not fully describe the various hypotheses they imply. I wish not to be too harsh on this point, because it is endemic in NLP and much of ML. However, in this case it is especially important, because the authors are introducing a number of hypotheses: (a) Unsupervised neural representations from BERT can be disentangled into constituent representations that stand for different time scales. (b) This disentanglement can happen at the level of the *neuron*. Note that (a) does not imply this, because it could be that some transformation of the information that requires global knowledge of the given vector is required for disentanglement, i.e. rotation. (c) The tasks chosen to represent the timescales actually do operate on those timescales.

Correctness: Figure 4 convinces to me that BERT’s representations can be at least partially timescale disentangled on the neuronal level, though citing previous work to show that these tasks operate at different timescales is the bare minimum of what should be required here. Past the bare minimum, experimenting on more datasets would make these results much stronger. Of course this is true for almost every paper, but since this paper is suggesting that (a) there is unsupervised timescale structure in BERT representations (b) the chosen tasks primarily require information from the desired timescale and (c) the evidence for (a) is (b), we need to be very convinced that the tasks correlate with timescales in order to be sure that these results mean what we think they mean.

Clarity: The paper is relatively clear, though as suggested in “Weaknesses” it hides some of its assumptions.

Relation to Prior Work: The description of the related work is adequate for putting the paper in context.

Reproducibility: Yes

Additional Feedback:


Review 2

Summary and Contributions: The paper applies spectral filtering to contextualized word embedding. It observes that the outputs at different frequency bands are responsible for various down-steam tasks that depend on different context length. For example, low-pass filtered output is responsible for topic classification, which requires a long context. The authors then propose a method that transforms word embedding dimensions with different band-pass filters. The method improves performance on tasks that require longer context.

Strengths: The paper provides an interesting perspective to the question "how contextualized are contextualized word embeddings". It implicitly shows that BERT learned embeddings cannot encode very long-range context. The message is interesting, and will encourage people re-thinking the success of BERT. In addition, the Fourier analysis is simple yet elegant.

Weaknesses: The Prism layer can be considered as applying a group of linear transforms to the embedding matrix. These transforms are fixed, corresponding to different frequency response. How does it compare with a learned transform?

Correctness: No obvious mistakes are found.

Clarity: The paper is well written.

Relation to Prior Work: Adequate

Reproducibility: Yes

Additional Feedback: In fig.5, why BERT+prism is worse for index outside [200, 300]? How does the Fourier coefficients look like at each frequency band? Visualizing them will be very interesting.


Review 3

Summary and Contributions: This original paper proposes to use tools from spectral filtering to specialize the output of a NLP language model to different _scales_ (intuitively word / phrase / document). The author first propose to apply those filter on top of a pre-trained language model (BERT) and show through probing (linear classifier) that the resulting embeddings indeed correspond to different hierarchical scales in NLP. In addition their approach can be integrated as a differentiable layer into the learning procedure, and the authors show interesting results confirming their hypothesis by obtaining improved performance of base-BERT. POS-REBUTTAL: Thanks a lot for the clarifications, and promises of more experiments. I would increase my score to 7.5 if I could based on those comments. I encourage the authors to perform those experiments, as it will strengthen this already good paper.

Strengths: - a nice approach to the problem of multi-scale representation of units which does not rely on a linguistic prior. A lot of work has been done into integrating such different levels in a hard-coded way (eg, by using char embeddings combined with word embeddings). This proposal has the advantage of not having to decide on those points a priori - Fig 4 which strongly shows that the retrieved bands indeed correspond to different levels of meaning - the possibility of integrating this during training opens up many new possibilities. While this is not a contribution of this paper, they take advantage of an implementation compatible with torch.

Weaknesses: - Sect 4 is rather weak. The experiments are very simplistic (result on GLUE would have been more exhaustive), and the precise choice on where to use the prism layer raises some questions (more on this below) - if accepted, I would hope the author reach to the NLP community, as researchers interested in probing would be very interested in this approach and might not discover this work

Correctness: Yes, the reasoning seems fine. The experiments could have been more exhaustive and solid, but they do run 3 runs and report standard deviation.

Clarity: Yes, the authors do not assume a lot of prior knowlege on spectral transformation and do a good job in explaining topics which might not be known by a majority of NeurIPS attendance

Relation to Prior Work: afaik this is very novel. I would have liked see some discussions with respect to the recent ACL position paper [1]. [1] The Unstoppable Rise of Computational Linguistics in Deep Learning. James Henderson. ACL 2020

Reproducibility: Yes

Additional Feedback: Why using the prism layer only at the last layer? Wouldn't it make sense to use in each intermediate layer? The way of dividing the embeddings into 5 sectors seems a bit naive. Transformer have multi-head attention, why not applying different prism for different attentions? It would then be up to the model to decide what to use from each head. In general, the architectural choice on where to include the prism layer seems to me the weakest part of this paper, and I would have liked to see more discussion and experiments on how to use it.


Review 4

Summary and Contributions: ===================== UPDATES TO THE REVIEW ================== I would like to thank the authors for taking the time to respond to my questions and concerns. Thank you for helping me better understand the reason for why performance on sentence/document classification is poor. I would nonetheless strongly encourage you to discuss this in more detail in your revisions and maybe include potential suggestions on how to get this to work well for such problems. ============================================================ The paper presents an approach to building multi-scale representations of text by filtering out high/low frequency information from a transformer's representations using DCT. The authors evaluate this setup using low, medium and high filters on BERT representations depending on the nature of the task (ex: POS tagging uses high filters while topic/document classification uses low filters).

Strengths: 1. The idea of using DCT to filter out low/high frequency information from the representations learned by pre-trained transformer LMs is novel and interesting. Although, this is a fairly common practice in computer vision to produce image representations. 2. The approach is straightforward and easy to use. 3. It is a general plug-and-pay model agnostic method to obtain multi-scale representations of text. 3.

Weaknesses: The biggest limitation of this work for me, is the experimental setup, specifically (1) the lack of comparison to existing models (2) poor results on text classification and speech act classification when compared to existing work and (3) the choice of benchmarks. - The only baseline approach compared against is BERT. I would recommend reporting results presented in previous work on POS tagging, speech act classification and text classification. This is particularly important since you run your own BERT baselines, it would be for the reader to know how these baselines compare with numbers reported in other papers. For example, [1] reports results on 20Newsgroups and [2,3] on the switchboard dialog act classification dataset and [4,5] on POS tagging. - I am curious why your classification accuracies on the 20newsgroups and switchboard datasets are much lower than what is reported in the literature? For example [1] reports 86.8% accuracy on 20newsgroups while you report only 32.21% for BERT and 51.01 for BERT + Prism. Similarly, [2] reports an accuracy of ~79% for BERT on the switchboard dataset while you report only ~47%. [3] reports a pretty strong non-BERT baseline of 82.9%. These differences are quite large and I wouldn't feel comfortable vouching for this paper unless your reported numbers are in this ballpark. As a sanity check, you could try to see what happens if you don't finetune the initial BERT model on wikitext-103? - Accuracy is not the typical evaluation metric used to evaluate POS-taggers in the literature - micro or macro averaged F1 is typically used. For example, see [4]. - Since Figure 5 demonstrates good performance on long range masked language modeling, LAMBADA might be a good benchmark to validate this. - It would be nice to see ablations where you use high filters on POS tagging and low filters on paragraph/document classification to see the gains that come from choosing the right set of filters for each task. [1] "Neural Attentive Bag-of-Entities Model for Text Classification" [2] "Deep Dialog Act Recognition using Multiple Token, Segment, and Context Information Representations" [3] "Dialogue Act Classification with Context-Aware Self-Attention" [4] "End-to-end Sequence Labeling via Bi-directional LSTM-CNNs-CRF"

Correctness: The overall claim of using DCTs to learn multi-scale representations is sound. However, claims that they improve model performance is not clear since the numbers reported on the experiments aren't comparable to existing work. It is also unclear to me if dialog act classification and text classification benchmarks should be labeled as "probing for linguistic information". There are probing tasks to evaluate certain linguistic phenomena in contextualized word representations ex: https://arxiv.org/abs/1901.05287 and the SentEval probing tasks - https://github.com/facebookresearch/SentEval.

Clarity: Yes, the paper is well written.

Relation to Prior Work: Yes, the prior work sufficiently discusses hierarchical and multi-scale models in NLP and similar spectral approaches in audio and computer vision.

Reproducibility: Yes

Additional Feedback:

[Author Response · NeurIPS 2020]

*General response to reviewers:* We thank the reviewers for their insightful and useful feedback! We are encouraged that they believe our work "opens up many possibilities" (**R3**), and is "very novel" (**R3**), "promising" (**R1**), "simple yet elegant" (**R2**) and "interesting" (**R2**, **R3**, **R4**). Reviewers appreciated that much prior work has involved hard-coding levels of linguistic structure into different model architectures (**R3**) and that our method is a "general plug-and-play model-agnostic method" (**R4**) that does not need to "decide on these points a priori" (**R3**).

*Larger points:* **R4**'s primary concern is the gap between the performance of our BERT + prism model and SOTA for the tasks we consider, and helpfully provides citations for SOTA models. However, the purpose of these experiments is to consider how tools from spectral analysis can affect the representations of a single, general-purpose model (BERT) across different tasks. This has both intrinsic scientific value as well as practical value, as these techniques could then be integrated into the various task-specific architectures. Two other factors explain this gap: 1) BERT contextual embeddings are known to perform surprisingly poorly on tasks beyond the word-level (see e.g. [2], where BERT embeddings perform up to 16 points worse than even GloVe embeddings) 2) SOTA models are trained end-to-end, while we train only a logistic regression on top of the token-level BERT embeddings without scale-specific pooling.

*Other points:* **R1** : **"The authors do not fully describe the various hypotheses they imply..."** This is a great point, and we'll make sure points 2(a), (b), and (c) that **R1** mentions are made explicit in the next revision.

**R2** : **Prism layer transforms are fixed; how does this compare to a learned transform?** This is an interesting question. When pretraining with the prism layer, a fixed transform is useful because the MLM task is very local (Fig. 4) and the fixed bands encourage the model to produce multiscale representations. Learning the transform to optimize the MLM task may lead to all the frequency bands becoming local. However, it is an interesting direction for future work whether clever regularization can get the best of both worlds here.

**R2** : **"In fig.5, why is BERT+prism worse for indices outside [200, 300]?"** We touch on this in L195-198, but the higher loss around the masked region suggests that the model relies more on the (redacted) distant context than BERT does to perform the MLM task, as we'd expect given the prism constraint.

**R3** : **"precise choice on where to use the prism layer raises some questions..."** We agree that it would be very interesting to explore other ways of using the prism layer, including between every layer. We will include a comparison to this setting for the camera ready.

**R3** : **"The way of dividing the embeddings into 5 sectors seems a bit naive"** We made this choice primarily to enable clear comparisons between tasks at different linguistic scales. In practice, one could choose particular scales relevant to desired end tasks or smoothly shift the frequency band between individual neurons as opposed to sectors of neurons. We will note this in the paper, and that there is opportunity for future work!

**R4** : **"It would be nice to see ablations where you use high filters on POS tagging and low filters on para-graph/document classification to see the gains that come from choosing the right set of filters for each task."** We thank **R4** for the useful suggestion. For the lowest frequency sector of the BERT + prism model, topic classification accuracy is 45.1% (vs 51.0 for the full model and 5.3% for the highest-frequency sector). With the highest-frequency sector, POS tagging accuracy is 84.1% (vs 94.4 for the full model and 16.8% for the lowest-frequency sector). This suggests that the bands are largely but not entirely responsible for the high performance of BERT + prism, as expected.

**R4** : **"As a sanity check, you could try to see what happens if you don't finetune the initial BERT model on wikitext-103?"** We thank **R4** for the useful suggestion. The original BERT model achieves an accuracy of 94.6% for POS tagging, 41.8 for dialog acts, 28.9 for topic classification, slightly worse than our model that was trained longer on WikiText-103 (95.9%, 47.1%, 32.2% respectively). As before, these are trained on contextualized token representations and are not comparable to models that perform pooling or are trained end-to-end. We will include this ablation in the next revision.

**R4** : **"Since Figure 5 demonstrates good performance on long range masked language modeling, LAMBADA might be a good benchmark to validate this."** This is an interesting idea for future work, as it is not yet straightforward to use BERT for assigning probabilities to multi-token words in LAMBADA given 1) the constraints of the masked language modeling interface and 2) that LAMBADA lacks end of sentence punctuation needed for bidirectional conditioning.

Reviewers also offered a number of other suggestions which we are grateful for and will incorporate into the final version of our paper.

*References:* [1] Understanding intermediate layers using linear classifier probes [2] Sentence-BERT: Sentence Embeddings using Siamese BERT-Networks

[Meta-Review · NeurIPS 2020]

After reading each other's reviews and discussing the author response, reviewers lean towards acceptance of this submission. The primary weaknesses are related to the lack of analysis in the fine-tuning setting and a resulting uncertainty about the use case of this technique in a practical sense. On the positive side, the analysis itself is informative and introduces a new angle to the work on multi-scale representations. I encourage authors to consider reviewer feedback for future revisions -- especially those from R1 about the assumptions implicit in the work.